# A Novel Mineral-like Copper Phosphate Chloride with a Disordered Guest Structure: Crystal Chemistry and Magnetic Properties

**DOI:** 10.3390/ma15041411

**Published:** 2022-02-14

**Authors:** Galina Kiriukhina, Olga Yakubovich, Larisa Shvanskaya, Anatoly Volkov, Olga Dimitrova, Sergey Simonov, Olga Volkova, Alexander Vasiliev

**Affiliations:** 1Faculty of Geology, Lomonosov Moscow State University, 119991 Moscow, Russia; g-biralo@yandex.ru (G.K.); yakubovich.olga320@gmail.com (O.Y.); lshvanskaya@mail.ru (L.S.); toljha@yandex.ru (A.V.); dimitrova@list.ru (O.D.); 2Institute of Experimental Mineralogy RAS, 142432 Chernogolovka, Russia; 3Quantum Functional Materials Laboratory, National University of Science and Technology “MISiS”, 119049 Moscow, Russia; os.volkova@yahoo.com; 4Institute of Solid State Physics RAS, 142432 Chernogolovka, Russia; simonov@issp.ac.ru; 5Faculty of Physics, Lomonosov Moscow State University, 119991 Moscow, Russia; 6Institute of Physics and Technology, Ural Federal University, 620002 Ekaterinburg, Russia

**Keywords:** crystal structure, low temperature, phosphates, magnetic properties, hydrothermal synthesis, crystal chemistry

## Abstract

Novel copper phosphate chloride has been obtained under middle-temperature hydrothermal conditions. Its crystal structure was established based on the low-temperature X-ray diffraction data: Na_2_Li_0.75_(Cs,K)_0.5_[Cu_5_(PO_4_)_4_Cl]·3.5(H_2_O,OH), sp. gr. *C*2/*m*, *a* = 19.3951(8) Å, *b* = 9.7627(3) Å, *c* = 9.7383(4) Å, *β* = 99.329(4)°, *T* = 150 K, Mo*K*_α_ (λ = 0.71073 Å), *R* = 0.049. The crystal structure includes tetrameric copper clusters as the main building blocks, which are built of four CuO_4_Cl pyramids sharing apical Cl vertices. The clusters are combined through phosphate groups and additional copper-centered polyhedra to form two mostly ordered periodic layers. Between the layers and inside the framework channels, alkali ions, H_2_O molecules, or OH groups are statistically distributed. Na_2_Li_0.75_(Cs,K)_0.5_[Cu_5_(PO_4_)_4_Cl]·3.5(H_2_O,OH) is a synthetic modification of a sampleite-polymorph of the lavendulan mineral group and represents a new member in a mero-plesiotype series of copper phosphates and arsenates, for which the crystal structures contain two-periodic [Cu_4_*X*(*T*O_4_)_4_]_∞_ modules (*T* = As, P; *X* = Cl, O). Magnetically, this phase exhibits the phase transition at *T_C_* = 6.5 K, below which it possesses a weak ferromagnetic moment.

## 1. Introduction

Open-framework materials comprise a wide range of crystalline compounds of different chemical classes [1,2]. Along with the zeolites comprised by more than 255 unique framework types [3], metal–organic frameworks, permeable polymers, and hybrid organic–inorganic framework materials, the inorganic metal phosphates, oxyfluorides, nitrides, and sulphides with three-periodic porous structures are well known [4,5]. The interest in these materials is due to their multiple applications in catalysis, gas and ion separations, ion exchange, gas adsorption and storage, waste treatment, anion recognition, etc. [6]. The amazing variety of such properties is related to their structure peculiarities: robust 3D framework structures enclose pores of different sizes that may comprise extra-framework molecules and atoms.

Among inorganic materials with an open framework, transition metal phosphates are scarce and mainly include Mo, V, Fe, Co, Mn, Zr, and Ti cations, and thus only three zeolites with the copper phosphate composition are known [3]. Presumably, metal cations prefer to be arranged in polyoxometalates (POMs) that are composed of oxometal polyhedra and polyanion clusters with structural diversity [7]. Among them, polyoxocuprates (POCus) have been distinguished, showing cross-structural topological transformations related to their multiple physical properties in homogeneous photocatalysis, medical chemistry, molecular magnetism, and quantum computing [8]. Recent discoveries of new rare and complex minerals demonstrate the presence of natural POMs and POCus in geochemical systems [9,10]. Here, we report the hydrothermal synthesis, crystal structure, and physical properties of new complex copper phosphate chloride, Na_2_Li_0.75_(Cs,K)_0.5_[Cu_5_(PO_4_)_4_Cl]·3.5(H_2_O,OH), which is a new member of the polysomatic series of the lavendulan mineral group. As seen from the complex formula, the new compound is capable of capturing and accumulating various alkali metal ions, from a small to large radii. It is promising to study its potential in the field of waste processing and ion-exchange technologies.

## 2. Materials and Methods

### 2.1. Hydrothermal Synthesis and Crystallization

It is well known that natural geothermal systems are multicomponent. In particular, typical alkali metal cations of chloride geothermal waters are Na^+^, Li^+^ and K^+^ [11]. Following the synthetic approach, the presence of various alkali metal cations impacts the crystallization process under hydrothermal conditions. For example, the presence of smaller Li^+^ can reduce the crystallization time [12], whereas the larger one, such as Cs^+^, plays a key role in the formation of microporous phases in the absence of organic directing agents [13,14]. To obtain mineral-like phases with open framework structures, we provided our exploratory synthesis in a system enriched with alkali metal salts. The main chemical agents were CsCl (0.5 g; 3 mmol), CuCl_2_ (1 g; 7.4 mmol), and Na_3_PO_4_ (1 g; 6.1 mmol). To simulate the composition of the natural geothermal medium, we added a small amount of KCl (0.1 g; 1.3 mmol) and LiCl (0.05 g; 1.2 mmol) to the initial mixture.

All reagents were stirred in an agate mortar, put into a 6 mL standard Teflon-lined stainless-steel pressure vessel, and filled with distilled water up to 83% of its volume. The autoclaves were heated to 523 K in a furnace, isothermed for 20 days (for completion of the reaction), and followed by cooling to room temperature for over 24 h. Crystallization products were washed with hot distilled water and dried at room temperature. Bright turquoise slim plates up to 0.2 mm were selected under a binocular microscope (Figure 1). Few green column crystals were found in close intergrowth with the main phase (Appendix A). According to the powder X-ray diffraction analysis (see Appendix A), this impurity crystalline phase was identified as Cu_3_(PO_4_)_2_ [15]. Chemical analysis was performed on a JEOL JSM-6480LV Oxford X-Max^N^ energy-dispersive diffraction spectrometer (JEOL, Akishima, Japan) at the Laboratory of Local Methods for Studying Materials (Department of Petrology, Faculty of Geology, Lomonosov Moscow State University). The elemental chemical composition based on EDS (JEOL, Akishima, Japan) was Na, Cs, K, Cu, P, Cl, and O atoms for the main phase.

### 2.2. X-ray Experiment and Crystal Structure Determination

Single crystal X-ray diffraction data were collected at *T* = 150 K using MoK_α_ radiation with an Oxford Diffraction Gemini single crystal diffractometer (Agilent Technologies, Yarnton, UK) equipped with a CCD detector; Mo *K*α radiation (*λ* = 0.71073 Å). The dataset was corrected for background, Lorentz and polarization effects, and absorption [16]. All of the calculations were performed within the WinGX program system [17]. The crystal structure was solved by direct methods and refined using SHELX programs [18,19]. Crystal data and details of data collection and refinement are presented in Table 1 and in the Appendix A–S3. The refined structural formula was (Na_0.98_Cu_0.02_)_2_Li_0.74_(Cs_0.33_K_0.20_H_2_O_0.12_)[Cu_5_(PO_4_)_4_Cl·H_2_O]Cl_0.07_·2.5(H_2_O,OH). 

The X-ray diffraction pattern at room temperature, photograph of crystals, fractional atomic coordinates and isotropic or equivalent isotropic displacement parameters, interatomic distances, and hydrogen-bond geometry for Na_2_Li_0.75_(Cs,K)_0.5_[Cu_5_(PO_4_)_4_Cl]·3.5(H_2_O,OH) are given in the Appendix A.

## 3. Results

### 3.1. Interatomic Distances and Crystal Structure Description

The basic structural units of Na_2_Li_0.75_(Cs,K)_0.5_[Cu_5_(PO_4_)_4_Cl]·3.5(H_2_O,OH) are shown in Figure 2a. Two phosphate tetrahedra are strongly distorted with P–O bond lengths ranging from 1.519(5) to 1.561(5) Å; the largest P–O1 and P–O2 distances correspond to bridging O atoms, which also participate in the first coordination spheres of two Cu atoms. Four symmetrically independent Cu atoms have four nearest O with Cu–O distances in the interval 1.938(5)–1.975(4) Å. The fifth O atom in the tetragonal-pyramidal coordination of Cu1 at a distance of 2.25 Å belongs to the H_2_O (O9) molecule. Cu2-, Cu3-, and Cu4-centered polyhedra share one Cl atom at a distance of about 2.70 Å, which also completes their coordination to a tetragonal pyramidal shape (Figure 2a). Therefore, Cu polyhedra possess Jahn–Teller distortion typical for Cu^2+^ cations.

The CuO_4_Cl pyramids share common edges and the Cl-vertex to form tetrameric clusters, which are additionally connected by orthophosphate tetrahedra in the [Cu_4_Cl(PO_4_)_4_]^5−^ blocks parallel to the *bc* plane (Figure 2b). On both sides of the blocks along the *x* axis, the Cu1O_4_(H_2_O) tetragonal pyramids are attached by vertex-bridge contacts with PO_4_ tetrahedra (Figure 3). Alkali cations Na^+^, Li^+^, Cs^+^, K^+^, and H_2_O molecules are located between the [Cu_5_(PO_4_)_4_Cl(H_2_O)]^3−^ slabs (Figure 3), and compensate for their negative charge. Sodium atoms occupy two positions at the centres of the tetragonal bicapped pyramid (Na1) (Figure 4a) and the trigonal prism (Na2) (Figure 4b) with Na–O distances ranging between 2.338(8) and 2.632(4) Å (average 2.42 Å). The occupancy factors for Na atoms are 98%; at a close distance from them, additional Cu atoms (Cu5 and Cu6) are distributed in the amount of 2%. Strongly distorted LiO_4_ tetrahedra, statistically populated by 74% Li atoms, are characterized by two Li–O distances of 1.76(1) Å and two significantly larger ones, equal to 2.44(2) Å (Figure 4c).

Large Cs^+^ and K^+^ ions, and H_2_O molecules, statistically occupy three neighbouring positions on the two-fold axis. The Cs^+^ fills 11-vertex polyhedron with bond lengths Cs–O in range 3.130(5)–3.452(4) Å and Cs–Cl1 equal to 3.553(3) Å, at 33% (Figure 4d). The K^+^ (S.O.F. 0.2) is located at a distance of 0.71(3) Å from Cs^+^ and is surrounded by O and Cl ligands with K–O and K–Cl bond lengths varying from 3.162(13) to 3.57(2) Å (Figure 4e). The H_2_O molecules are placed in the mirror plane (site symmetry 2/*m* statistically populated for 24%). Thus, the crystal structure of Na_2_Li_0.75_(Cs,K)_0.5_[Cu_5_(PO_4_)_4_Cl]·3.5(H_2_O,OH) features a “combination” of ordered two-periodic heteropolyhedral anionic blocks [Cu_5_(PO_4_)_4_Cl(H_2_O)]^3−^ that alternate with highly disordered layers of alkaline cations and water molecules. Interatomic distances for all polyhedra can be found in Appendix A, and hydrogen-bond geometry in Appendix A.

### 3.2. Crystal Chemistry

Na_2_Li_0.75_(Cs,K)_0.5_[Cu_5_(PO_4_)_4_Cl]·3.5(H_2_O,OH) represents one more member in the group of minerals and synthetic compounds interpreted as polysomatic series of crystal structures with two-periodic heteropolyhedral modules [Cu_4_*X*(*T*O_4_)_4_] built of tetrameric clusters of Cu-centred polyhedra, and phosphate or arsenate tetrahedra [20]. It is noteworthy that Cu-centred polyhedra are distorted in accordance with the Jahn–Teller effect; besides, they obey tetragonal symmetry, which tetrameric clusters inherit. Furthermore, the [Cu_4_*X*(*T*O_4_)_4_] slabs also feature tetragonality (Figure 2b). Accordingly, half of the series members crystallize with tetragonal symmetry, like mahnertite and its synthetic analogue (*I*4/*mmm*) [21], or the simplest isostructural phosphates Ba(VO)Cu_4_(PO_4_)_4_, Ba(TiO)Cu_4_(PO_4_)_4_, and Sr(TiO)Cu_4_(PO_4_)_4_ (*P*42_1_2), as well as Sr_2_Cu_5_(PO_4_)_4_Cl_2_·8H_2_O(*P*42_1_2) and Sr_2_Cu_5_(PO_4_)_4_Br_2_·8H_2_O (*P*4/*nmm*). Nevertheless, the rest of the compounds have a reduced orthorhombic symmetry (calicoandyrobertsite-2*O* and synthetic arsenates RbNa_5_Cu_4_(AsO_4_)_4_Cl_2_, CsNa_5_Cu_4_(AsO_4_)_4_Cl_2_) or monoclinic systems (seven minerals and our phase).

Similar modules *A* [Cu_4_*X*(*T*O_4_)_4_] in the crystal structures of copper phosphates and arsenates alternate with the second-type modules *B*, which vary in composition and topology. According to the classification by Makovicky, the series is merotypic [22]. At the same time, all members are built from modules *A* slightly differing in chemistry and configuration; this fact denotes the series as plesiotype [22,23]. Considering both characteristics, the series of compounds under discussion should be called mero-plesiotype.

The presence of similar (pseudo)tetragonal slabs [Cu_4_*X*(*T*O_4_)_4_], assembled from (CuO_3_)_4_Cl/(CuO_3_)_4_O clusters and PO_4_/AsO_4_ tetrahedra, underlies the “equality” of two unit-cell parameters for all phases (Figure 2b). For example, *b* and *c* parameters for Na_2_Li_0.75_(Cs,K)_0.5_Cu_5_(PO_4_)_4_Cl·3.5(H_2_O,OH) are both close to 9.7 Å, which is distinctive for phosphate members. For arsenates, these values are normally larger and reach 10.0 Å. For richelsdorfite and two synthetic isostructural compounds, RbNa_5_Cu_4_(AsO_4_)_4_Cl_2_ and CsNa_5_Cu_4_(AsO_4_)_4_Cl_2_, two pseudo-tetragonal parameters are close to 14 Å, which exactly corresponds to the diagonal directions of the 10 Å cell mentioned above. The third parameter of the unit cell varies significantly for the series members from 7.1 Å to 23.7 Å due to the different ways of assembling the two modules *A* and *B*.

The title crystal structure, apparently, is closely related to the structure of the polymorphic modification of the mineral sampleite, NaCaCu_5_(PO_4_)_4_Cl·4.5H_2_O [24]. It is very likely that in both structures, two types of modules alternate in the same sequence (*ABA’B’*…), where *A’* and *B’* are the inverted blocks *A* and *B*. This leads to the proximity of their third unit cell parameter being equal to ~19.5 Å. For both structures, the monoclinic angles take a value of 99°–102°, which is close to the *β* angle in some other minerals—richelsdorfite, (calico)andyrobertsite, and epifanovite. All other compounds of this polysomatic series exhibit an angle close to 90°, which is obviously related to a strong (pseudo)tetragonality of their structures. 

The main feature of our phase is the presence of exclusively alkali monovalent cations (even of four types), which form the *B* module, while the other members, including sampleite, always contain divalent alkali-earth elements like Ca, Sr, Ba, Cd, or even Pb. Using four different alkaline cations in our hydrothermal experiment, we tried to simulate a natural hydrothermal process. As a result, the monoclinic Na_2_Li_0.75_(Cs,K)_0.5_Cu_5_(PO_4_)_4_Cl·3.5(H_2_O,OH) phase was obtained similarly to most monoclinic natural minerals, but was the first case among all synthetic tetragonal or orthorhombic compounds of the series under discussion.

Among natural and synthetic inorganic compounds, the recent review mentions at least 26 types of CuO*_m_*Cl*_n_* coordination polyhedra with O and Cl ligands [25]. While the [4O+Cl] square pyramid is a typical Cu^2+^-centered polyhedron with Jahn–Teller distortion, a tetrameric cluster of four copper pyramids sharing a common chlorine vertex turned out to be rather rare. As described above, the [(CuO_3_)_4_Cl] clusters (Figure 5) are basic structural units for our compound and other members of the lavendulan polysomatic series, which includes secondary arsenate and phosphate copper minerals. Note, the tetrameric clusters of [(CuO_3_)_4_Cl] were recently recognized in the synthetic phase Na_6_Cu_7_BiO_4_(PO_4_)_4_[Cl,(OH)]_3_. In its composite crystal structure, the copper tetramers form layers of square-kagomé topology responsible for a possible quantum spin-liquid property [26].

### 3.3. Physical Properties

The temperature dependences of the thermodynamic properties, i.e., magnetization *M* and heat capacity *C*_p_, in the range 2–300 K under magnetic field *B* up to 9 T, have been measured on the pressed pellets of crushed tiny crystals using relevant options of the “Quantum Design” Physical Properties Measurement System PPMS-9T. 

The temperature dependences of the magnetic susceptibility *χ* = *M*/*B* in Na_2_Li_0.75_(Cs,K)_0.5_[Cu_5_(PO_4_)_4_Cl]·3.5(H_2_O,OH) measured in the field-cooled (FC) and zero-field-cooled regimes at *B* = 0.1 T are shown in Figure 6. At high temperatures, these curves coincide within experimental error, following the Curie–Weiss law. The fitting in the range 100–300 K gives the Weiss temperature, Θ = −25.7 K, pointing to the predominance of the antiferromagnetic interactions in the system. The Curie constant *C* = 2.27 emu K/mol corresponds to the presence of five Cu^2+^ ions with spin S = 1/2 and g-factor g = 2.2. With lowering the temperature, the experimental data deviated downward from the extrapolation of the Curie–Weiss law, which indicates the predominance of the antiferromagnetic correlations in the system. At *T**~22 K, a weakly pronounced anomaly is observed in both FC and ZFC curves ascribed to the presence of the Cu_3_(PO_4_)_2_ secondary phase, as evidenced by X-ray diffraction [27]. This was followed by a sharp upturn at *T_C_* = 6.5 K in the FC curve and a downward inclination in the ZFC curve. The sharp dichotomy of FC and ZFC curves at *T* < *T_C_* indicates the presence of a weak ferromagnetic component and/or disorder in the magnetic subsystem of the low temperature phase.

Main magnetic units in the structure of Na_2_Li_0.75_(Cs,K)_0.5_[Cu_5_(PO_4_)_4_Cl] × 3.5(H_2_O, OH) are Cu_5_ pentamers organized by four edge-sharing CuO_4_Cl pyramids (Figure 5), which are connected via four phosphate groups with the fifth CuO_5_ pyramid, as shown in the inset to Figure 6. In the basal plane, the angle of Cu–O–Cu bond is equal to 107^o^, which corresponds to an antiferromagnetic exchange interaction *J* between the nearest neighbor basal centers in accordance with the Goodenough–Kanamori–Anderson rules [28]. The magnetic exchange interaction between basal and apical ions passes through the intermediate PO_4_ groups and is also usually antiferromagnetic *J′* [29]. The fit of *χ*(*T*) curve in the model of *S* = 1/2 pentamers can be done based on the Hamiltonian with exchange interaction *J* between two spins expressed as *Ĥ* = −2*JS*_1_·*S*_2_ [30]. Experimental values of the product of *χT* vs. *T* and the fit are shown in the inset of Figure 6. The parameters of magnetic exchange interactions are *J* = −19 K and *J′* = −24 K for g—factor g = 2.27 close to that obtained *χ*(*T*) curve.

The temperature dependence of the specific heat *C_p_* of Na_2_Li_0.75_(Cs,K)_0.5_[Cu_5_(PO_4_)_4_Cl] ·3.5(H_2_O,OH) measured in the quasi-adiabatic regime is shown in Figure 7. At about 200 K, *C_p_* is still far from the Dulong–Petit thermodynamic limit of about 10^3^ J/mol K. At *T*_C_, a pronounced anomaly in *C_p_*(*T*) is seen, corresponding to the phase transition in the magnetic subsystem. An additional anomaly at *T**~22 K corresponds to the ordering of the secondary phase Cu_3_(PO_4_)_2_ [27]. The lower inset shows the results of the specific heat measurements at several magnetic fields in the range from 0 to 9 T. Under a magnetic field, the phase transition at *T_C_* shifts upward in temperature at a rate of 0.085 K/T. An increase in the stability region of the low-temperature phase with magnetic field correlates with the presence of a weak ferromagnetic component in this phase.

## 4. Discussion

The studied Na_2_Li_0.75_(Cs,K)_0.5_[Cu_5_(PO_4_)_4_Cl]·3.5(H_2_O,OH) is structurally related to another complex copper phosphate Na_3_[Cu_5_(PO_4_)_4_F]·4H_2_O [31]. Both systems contain two-dimensional layers of strongly coupled Cu_5_(PO_4_)_4_F or Cu_5_(PO_4_)_4_Cl units, as shown in Figure 8. However, magnetic subsystems demonstrate different properties under the variation of halogen. The fluorine compound demonstrates a broad hump at 19 K and antiferromagnetic phase transition at *T_N_* = 11.2 K. The obtained parameters of basal and apical magnetic exchange interactions in Na_3_[Cu_5_(PO_4_)_4_F]·4H_2_O were determined as *J* = −24 K and *J′* = −14 K. Large difference between *J* and *J′* led to the manifestation first of the plaquette properties at *T* > *T*_N_, while in the studied chlorine compound, *J* and *J′* were closer, which made it possible to see the behavior of the copper pentamers. 

## 5. Conclusions

Copper phosphate chloride is synthesized in the form of single crystals under middle-temperature hydrothermal conditions. Low-temperature X-ray diffraction study made it possible to refine the complex and highly disordered crystal structures of Na_2_Li_0.75_(Cs,K)_0.5_[Cu_5_(PO_4_)_4_Cl]·3.5(H_2_O,OH), which is a novel synthetic modification of the sampleite-polymorph of the lavendulan mineral group and a new member in the mero- plesiotype series of copper phosphates and arsenates. The unique feature of its monoclinic layered structure is the possibility to accommodate various alkali ions between ordered copper-phosphate blocks.

## Figures and Tables

**Figure 1 materials-15-01411-f001:**
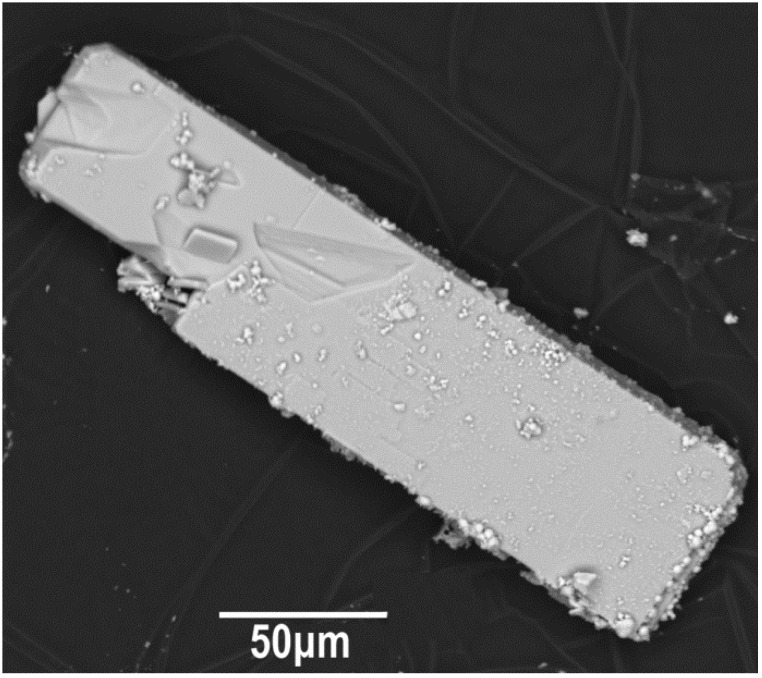
SEM images of the Na_2_Li_0.75_(Cs,K)_0.5_[Cu_5_(PO_4_)_4_Cl]·3.5(H_2_O,OH) phase.

**Figure 2 materials-15-01411-f002:**
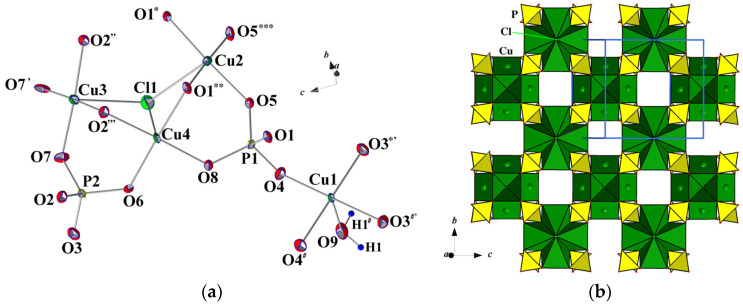
Basic structural units shown in ellipsoid mode at a 90% probability level (**a**). Symmetry codes: (′) *x*, 2 − *y*, *z*; (″) 1.5 − *x*, 0.5 + *y*, 2 − *z*; (‴) 1.5 − *x*, 1.5 − *y*, 2 − *z*; (*) 1.5 − *x*, 0.5 + *y*, 1 − *z*; (**) 1.5 − *x*, 1.5 − *y*, 1 − *z*; (***) *x*, 2 − *y*, *z*; (#) *x*, 1 − *y*, *z*; (*′) *x*, *y*, −1 + *z*; (#′) *x*, *y*, −1 + *z*. The ordered tetragonal [Cu_4_Cl(PO_4_)_4_]^5−^_∝_ block built from tetrameric Cu-units and phosphate tetrahedra (**b**).

**Figure 3 materials-15-01411-f003:**
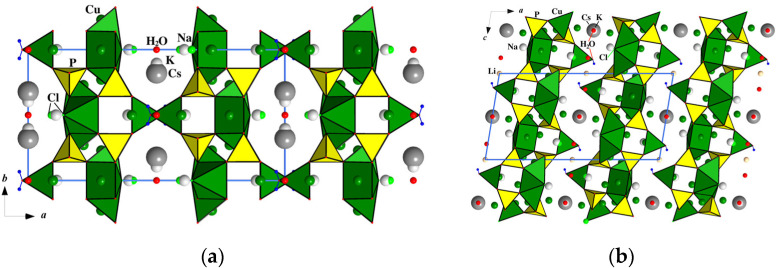
The crystal structure of Na_2_Li_0.75_(Cs,K)_0.5_[Cu_5_(PO_4_)_4_Cl]·3.5(H_2_O,OH) projected onto *xy* (**a**) and *xz* (**b**) planes showing an alternation of ordered Cu,P-blocks and highly disordered layers of alkaline metals and water molecules.

**Figure 4 materials-15-01411-f004:**
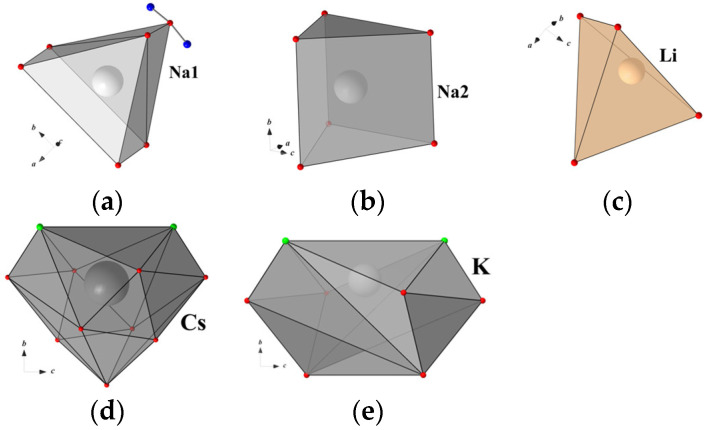
Coordination polyhedra around the alkali atoms in the title crystal structure: (**a**,**b**)-Na, (**c**)-Li, (**d**)-Cs, (**e**)-K.

**Figure 5 materials-15-01411-f005:**
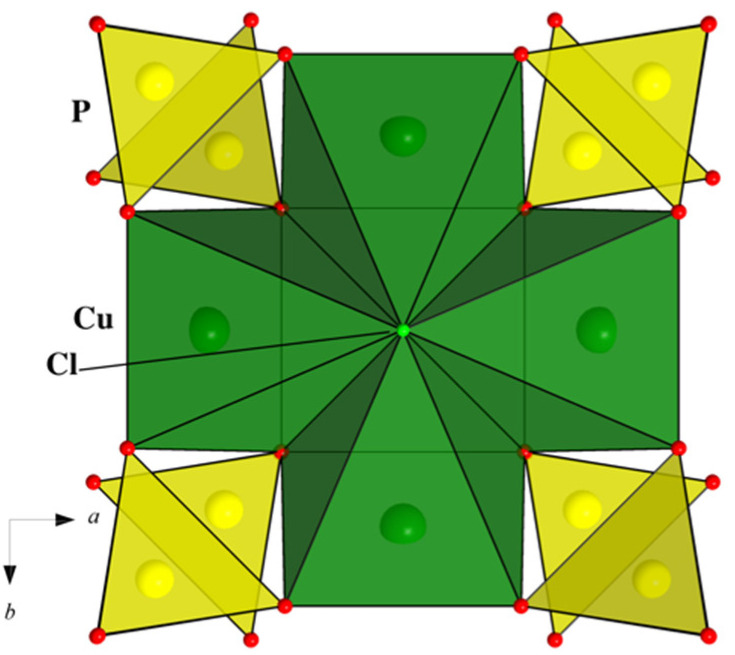
Clusters of copper tetramers and adjacent phosphate tetrahedra in the crystal structure of Na_2_Li_0.75_(Cs,K)_0.5_[Cu_5_(PO_4_)_4_Cl]·3.5(H_2_O,OH).

**Figure 6 materials-15-01411-f006:**
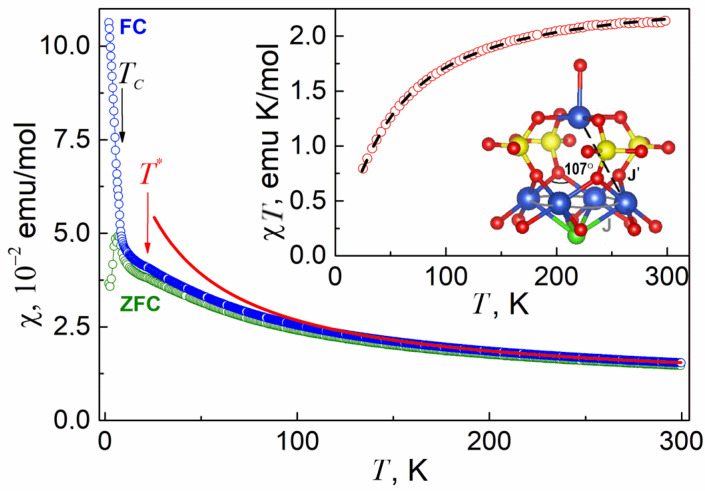
Temperature dependences of the magnetic susceptibility *χ* = *M*/*B* of Na_2_Li_0.75_(Cs,K)_0.5_[Cu_5_(PO_4_)_4_Cl]·3.5(H_2_O,OH) taken in both FC and ZFC regimes at *B* = 0.1 T. The solid line represents the extrapolation of the Curie–Weiss law at low temperatures. Arrow at *T*_C_ marks the magnetic phase transition. The *T** marks the impurity phase transition temperature. The inset: temperature dependence of the product *χT* and arrangement of Cu_5_ unit. *J* and *J**′* denote magnetic exchange interaction pathways. The dashed line is a fit with *S* = 1/2 pentamer model.

**Figure 7 materials-15-01411-f007:**
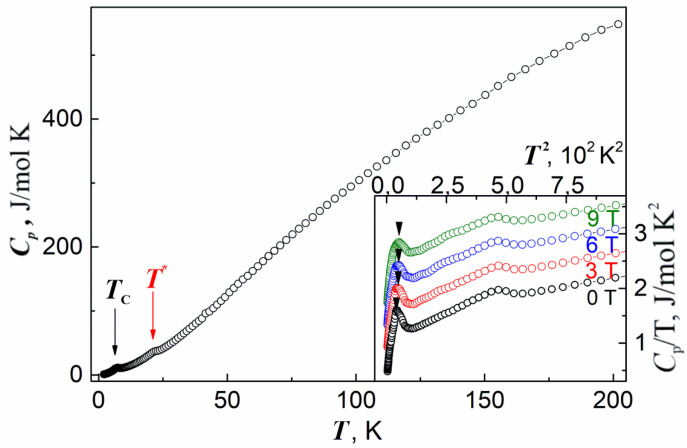
Temperature dependence of the specific heat *C*_p_ in Na_2_Li_0.75_(Cs,K)_0.5_[Cu_5_(PO_4_)_4_Cl]·3.5(H_2_O,OH). The inset: *C_p_*/*T* vs. *T*^2^ curves taken under magnetic fields of 0, 3, 6, and 9 T. The subsequent curves are shifted with respect to each other by increments of 0.4 J/mol K^2^. Arrows mark the magnetic phase transitions at *T*_C_ and the ordering of the secondary phase in T*.

**Figure 8 materials-15-01411-f008:**
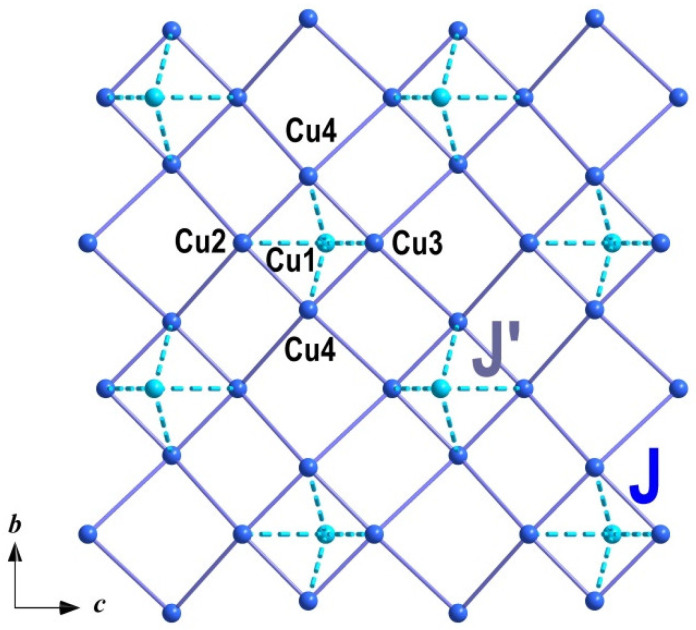
The geometry of magnetic network in Na_2_Li_0.75_(Cs,K)_0.5_[Cu_5_(PO_4_)_4_Cl]·3.5(H_2_O,OH).

**Table 1 materials-15-01411-t001:** Experimental details.

**Crystal Data**	
Chemical formula	Na_2_Li_0.75_(Cs,K)_0.5_[Cu_5_(PO_4_)_4_Cl]·3.5(H_2_O,OH)
M_r_	900.37
Crystal system, space group, *Z*	Monoclinic, *C*2/*m*, 4
Temperature (K)	150
*a*, *b*, *c* (Å)	19.3951 (8), 9.7627 (3), 9.7383 (4)
*β* (°)	99.329 (4)
*V* (Å^3^)	1819.54 (12)
Radiation type	Mo Kα (λ = 0.71073 Å)
µ (mm^−1^)	7.16
Crystal size (mm)	0.13 × 0.07 × 0.03
**Data Collection**	
Diffractometer	Xcalibur, AtlasS2, Gemini
Absorption correction	Analytical ^1^
*T*_min_, *T*_max_	0.997, 0.999
No. of measured, independent and observed [*I* > 2σ(*I*)] reflections	4570, 2219, 2066
*R* _int_	0.023
(sin θ/λ)max (Å^−1^)	0.650
**Refinement**	
*R*[*F*^2^ > 2σ(*F*^2^)], *wR*(*F*^2^), *S*	0.049, 0.103, 1.33
No. of reflections	2219
No. of parameters	179
No. of restraints	1
H-atom treatment	All H-atom parameters refined
Weighting scheme	*w* = 1/[σ2(*F*_o_^2^) + (0.010*P*)2 + 50.*P*] where *P* = (*F*o2 + 2*F*_c_^2^)/3
Δρ_max_, Δρ_min_ (e Å^−3^)	1.70, −0.93

^1^ Analytical numeric absorption correction using a multifaceted crystal model based on expressions derived by R.C. Clark and J.S. Reid [20]. Empirical absorption correction using spherical harmonics, implemented in the SCALE3 ABSPACK scaling algorithm.

## Data Availability

CCDC-2119350 contains the supplementary crystallographic data of title compound. These data can be obtained free of charge via www.ccdc.cam.ac.uk/data_request/cif, or by emailing data_request@ccdc.cam.ac.uk, or by contacting The Cambridge Crystallographic Data Centre, 12 Union Road, Cambridge CB2 1EZ, UK; fax: +44 1223 336033.

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
