# Peer review of "A Novel Mineral-like Copper Phosphate Chloride with a Disordered Guest Structure: Crystal Chemistry and Magnetic Properties"

_materials, 2022, doi:10.3390/ma15041411_

Round 1

Reviewer 1 Report

The authors have clarified interesting physical properties such as magnetism in relation to the crystal structure of Novel copper phosphate chloride.

This paper is judged to be worthy of publication in "Materials".

However, there are many notational mistakes, such as the fact that the manuscript does not contain what should be written in italics, so careful confirmation and correction by the authors is required for publication. 

Author Response

Response to Reviewer 1 Comments

The authors have clarified interesting physical properties such as magnetism in relation to the crystal structure of Novel copper phosphate chloride.

This paper is judged to be worthy of publication in "Materials".

However, there are many notational mistakes, such as the fact that the manuscript does not contain what should be written in italics, so careful confirmation and correction by the authors is required for publication. 

We are grateful to Reviewer for the positive estimation of our manuscript.

We have revised the text, notational mistakes were corrected

Author Response

Response to Reviewer 2 Comments

We are grateful to Reviewer for the thorough reading of our manuscript. 

- line 39 – 3D? corrected
- Line 76 – maybe the symbol ‡ can be changed from Appendix note and included as info in the text included in the text
- line 76 – delete space between “Elemental chemical” deleted
- Figure1 – please improve the scale image improved
- Table 1 – correct -1 as superscript in    corrected
- line 95 - correct X-ray    done
- Figure 8 – improve the quality of the image          improved

Reviewer 3 Report

The authors synthesized for the first time a novel copper phosphate chloride {(Na2Li0.75(Cs,K)0.5[Cu5(PO4)4Cl]*3.5(H2O,OH)} via a relatively simple hydrothermal treatment. They established its crystal structure by using low-temperature X-ray diffraction data, and characterized it from a magnetic point of view. The results are novel and of potential interest for the scientific community, thus I recommend the submitted manuscript for its publication in Materials.

However, the Introduction section could be improved by including additional hints on potential applications of the novel synthetic mineral.

Author Response

Response to Reviewer 3 Comments

The authors synthesized for the first time a novel copper phosphate chloride {(Na2Li0.75(Cs,K)0.5[Cu5(PO4)4Cl]*3.5(H2O,OH)} via a relatively simple hydrothermal treatment. They established its crystal structure by using low-temperature X-ray diffraction data, and characterized it from a magnetic point of view. The results are novel and of potential interest for the scientific community, thus I recommend the submitted manuscript for its publication in Materials.

However, the Introduction section could be improved by including additional hints on potential applications of the novel synthetic mineral.

We are grateful to Reviewer for the positive estimation of our manuscript 

We added in the introduction: "As seen from the complex formula, the new compound is capable to capture and accumulate various alkali metal ions, from small to large radii. It is promising to study its potential in the field of waste processing and ion-exchange technologies."